# *Streptococcus suis*: A Possible Emerging Zoonotic Pathogen in Romania

**DOI:** 10.3390/microorganisms13020335

**Published:** 2025-02-04

**Authors:** Serban Nicolae Benea, Ruxandra Moroti, Teodora Deaconu, Corina Ciont, Mihaela Anca Benea, Ilinca Savulescu Fiedler

**Affiliations:** 1Department of Infectious Diseases, “Carol Davila” University of Medicine and Pharmacy, 050474 Bucharest, Romania; serban.benea@umfcd.ro; 2National Institute for Infectious Diseases “Prof. Dr. Matei Bals”, 021105 Bucharest, Romania; gadeateodora@yahoo.com; 3“Marius Nasta” Institute of Pneumology, 050159 Bucharest, Romania; iaba.corina@yahoo.com; 4Clinical Hospital of Infectious and Tropical Diseases “Dr. Victor Babes”, 030303 Bucharest, Romania; anca_malin@yahoo.com; 5Department of Internal Medicine, “Carol Davila” University of Medicine and Pharmacy, 050474 Bucharest, Romania; 6Coltea Clinical Hospital, 030167 Bucharest, Romania

**Keywords:** *Streptococcus suis*, meningitis, deafness, ataxia, sequelae, pigs and pork, prolonged course, Europe

## Abstract

*Streptococcus suis* is a common germ in pig populations, with high carrier rates. Recent studies identify it as an emerging zoonotic pathogen, particularly in Southeast Asia, where raw pork is traditionally consumed. Data on *Streptococcus suis* infection in Europe, especially Eastern European countries like Romania, are limited. Our study reviewed data from an infectious diseases tertiary hospital in Bucharest between 2001 and 2024, including eight patients diagnosed with a *Streptococcus suis* invasive infection. The median age was 53.3 years, with a male-to-female ratio 3:1. Three patients had risk factors such as contact with pigs or handling fresh pork. Seven patients were initially diagnosed with meningitis and one with endocarditis. During hospitalization, an additional endocarditis case was identified among the meningitis patients. Laboratory samples indicated bacterial infection, with *Streptococcus suis* isolated from CSF in six cases and blood cultures in two cases. All strains tested were susceptible to beta-lactam antibiotics, but resistant to lincosamides and macrolides. There were no deaths, but half of our patients experienced severe meningitis-related sequelae, mainly hearing loss. Clinicians should be aware of *Streptococcus suis* as an etiologic agent of meningitis in non-endemic areas like Romania, especially in patients with risk factors (contact with pigs, pork).

## 1. Introduction

*Streptococcus suis* is a facultative anaerobic, non-motile Gram-positive coccus belonging to Streptococcaceae, order Lactobacillales, phylum Firmicutes. It represents both a serious human pathogen [1,2] and one of the most important bacterial pathogens affecting pigs, posing a significant health problem in the porcine industry worldwide [1].

*Streptococcus suis* is classified based on different antigenic reactions directed against the capsular polysaccharides. Based on the antigenic reaction, 35 serotypes were originally reported (serotypes 1–34 and serotype ½, which was named due to its reaction with both serotypes 1 and 2 antisera [1,3,4,5,6,7,8,9,10,11,12]. The last six serotypes were assigned to other bacterial species [3,12]; thus, the bacteria are now classified into 29 serotypes [1,13,14]. *Streptococcus suis* strains are classified furthermore based on sequence types determined through multilocus sequence typing. These sequence types are grouped into clonal complexes (CC_s_) [15], and studies have shown that CC1, which belongs to serotype 2 based on the capsule antigenic reaction, is the main strain causing infections. This lineage has been isolated from outbreaks and sporadic cases in all parts of the world [1]. Strains of serotype 2 and serotype 9 are frequently isolated from diseased pigs in Europe [3,5,16,17,18]. Some authors even proposed a link between the capsule of serotype 2 and the capacity of these strains to become harmful to humans and lead to zoonotic events [19,20].

**In pigs**, *Streptococcus suis* is a normal inhabitant of the upper respiratory tract, although it was also isolated from the reproductive and gastrointestinal tract of pigs. The carrier rate in pigs is very high, up to 100%, especially with nonvirulent serotypes. The real incidence of infection in the pig population is around 5% at any given time. It causes mainly meningitis, arthritis, and endocarditis and can lead to sepsis and sudden death; mortality rates can reach as high as 20%, especially in the young population [21,22,23].

Pig raising and pork production have represented an important social and economic tradition for a long time in Europe. For example, data show that in 2022, pig meat production in the EU represented 22% of total production worldwide, second after that in China (50–55%) but higher than in the United States (11%) [24,25]. Regarding pig meat consumption, the EU is the most important consumer, with 41.1 kg/per capita, followed by China with 37.2, the US with 29.5, and Russia with 24.1 kg/per capita [24,25].

Pig production is important in the European agricultural economy and represented 9% of the total agricultural production and 35% of total EU meat production in 2018 [25]. The main European zone for pig production extends in Western Europe, from Denmark—which holds the record for the number of pigs per capita, to Belgium and Spain. Traditionally, three major countries—Spain, France, and Germany—produce more than 50% of pork in Europe [26].

The production systems, farm size, and production methods vary widely in the EU, from Western Europe, with industrial installations and enormous farms keeping thousands of animals, to the East with small traditional farms that house between one and two pigs for their own consumption [27]. Nevertheless, keeping pigs in small farms usually means a high risk of infectious diseases, including a *Streptococcus suis* infection, and sanitation.

Romania is a country in East-Central Europe, near the Black Sea, covering a geographical area of 238 398 square kilometers. It has a population of 19,064,409 persons (data from the latest census) with a current population density of 82.29 people per square kilometer [28]. Romania has been a full member of the European Union since the 1st of January 2007, and its economy is considered a developing high-income mixed one [29]. Since becoming a full EU member, the country’s economic growth has maintained as one of the highest in Europe [30]. Pig raising and pork production are highly polarized in Romania. A report from the European Parliament shows that in 2015, Romania had more than half of the 2.2 million pig farms in the EU. Almost all farms (99%) had fewer than 10 pigs, most of them raising one or two pigs in a traditional, cheap growing system. All these backyard pig farms produce pork and products for home consumption or, in the best case, the local market. These farms house half of Romania’s pig herd. On the other hand, there are large intensive farms that supply more than 85% of all pork available in the country [31,32]. The African swine fever has hit the Romanian pig herds hard since the first case was diagnosed in 2017. Since then, almost 4000 outbreaks have been reported in the entire country, especially in the backyard system—as an example, in 2022, almost 90% of the total outbreaks of African swine fever (*n* = 327) in Europe appeared in Romania [33]. Rapidly implementing control measures was rather difficult because of this backyard growing system [34,35,36].

Although the carrier rate is so high in pigs, little data have been published from Romania. A quick search on PubMed on “Streptococcus suis” and “Romania” found only two studies, both published after 2023 [37,38]. Costinar et al. looked at the antimicrobial susceptibility of 267 strains of *Streptococcus* spp., including *Streptococcus suis* (*n* = 181, 67.79%), and reported high rates of resistance of *Streptococcus suis* to tetracyclines (87.23%), lincosamides (95.00%), and macrolides (72.22%). Almost 34% of the strains displayed resistance to penicillin. The authors also found high recovery rates of *Streptococcus suis* from the lungs and brains of diseased pigs [37]. Siteavu et al. also reported high rates of resistance for streptomycin (98%), tetracycline (75%), oxytetracycline (72%), doxycycline (52%), and erythromycin (51%) [38]. Two other studies, published in Romanian veterinary journals, found high resistance rates to the same classes of antibiotics [39] and identified *Streptococcus suis* infection as the main mortality cause in young piglets (68–70%) [40]. No data about circulating serotypes are available in Romania.

**In humans**, *Streptococcus suis* causes severe diseases, mainly meningitis and sepsis, but also endocarditis, arthritis, pneumonia, bacteremia, and cutaneous lesions. Hearing loss in humans could also be related to *S. suis* infection [1,2]. There are also reports of streptococcal toxic shock-like syndrome—an invasive infection of deep tissues, usually associated with producing bacterial super-antigens [2].

Since its first description in 1968, in Denmark, with two meningitis cases and one fatal sepsis [41], more than 1600 human cases of *Streptococcus suis* infection have been reported worldwide. Most cases originated from Asia (90.2%), notably more than in Europe (8.5%) and North America (0.5%) [3]. However, there are controversies regarding the real number of cases worldwide. For example, a review by Kerdsin et al. published in 2022 documented reports of cases of *Streptococcus suis* infection from Thailand between 1987 and 2021 and identified 1798 patients, more than the global number of cases reported by other sources [42]. To date, Thailand and Vietnam are considered endemic areas, and all Southeast Asia encounters significant health problems related to *S. suis* infections [42,43].

Human infections are linked with direct contact with the infected swine population and/or exposure to contaminated pork products, including consumption of undercooked contaminated pork [1,2,44,45]. Most studies, including a meta-analysis by Rayanakorn et al., have shown that the most important risk factors for developing *Streptococcus suis* disease are adult age, male sex, alcohol drinking, pig-related occupation or pig exposure, and consumption of raw or undercooked pork products, including blood [46,47]. Other reviews identified consuming raw pork/pig’s blood dishes as a risk factor for 61.4% of cases [42].

Some nuances are related to cultural behaviors in different parts of the world. Consumption of raw or undercooked pork, blood, and offal products is one of the main factors that lead to the increased incidence of human disease caused by *Streptococcus suis* in Southeast Asia [48,49,50,51,52]. The high density of pigs in this area represents another contributing factor. At this moment, Asia, especially China, is the world’s main region for pig production and pork consumption, concentrating almost two-thirds of the pig population in the world [52,53]. Another characteristic of the region is the fact that most pigs are raised on small-scale farms [54,55]. The number of human cases of *Streptococcus suis* infection has steadily increased in the past years, and Southeast Asian countries have reported the largest number of cases and outbreaks [3,11,42,43]. There are two main patterns when talking about *Streptococcus suis* disease in humans. On the one hand, there is Asia, especially its Southeastern part, with a high number of cases, and even outbreaks like those in China—in 1998 and 2005 [56]—and Thailand in March 2021 [57]. In the outbreak in Sichuan, China, in 2005, 215 cases were reported from mid-July to the end of August, and all infections occurred among farmers. Sixty-one (28%) of the patients had streptococcal toxic shock syndrome, and 62% (*n* = 38) of them died [56]. On the other hand, there are other countries with sporadic cases. In Europe and North America, including the United States, the estimated cumulative prevalence of human *Streptococcus suis* disease is lower than in Asia [1], and the cases are usually selected among occupationally exposed groups—slaughterhouse workers, butchers, and pig breeders [52]. In addition, most European studies found that skin lesions represent the point of entry for *Streptococcus suis* for those handling pigs or pork [3,52,58,59,60,61,62,63]. Besides skin contact, while handling or eating pork, direct contact with the mucosa of animals (pets, livestock, or even wild animals) may represent another way of acquiring the infection [50,64,65].

To date, infections caused by Streptococcus suis do not represent a notifiable disease in Europe, and there are no published guidelines for reporting such infections in a centralized manner. For this reason, *Streptococcus suis* infection is underdiagnosed, and the correct management and treatment are often delayed, increasing the risk of severe disease and lifelong sequelae.

Our study aims to raise awareness about this human pathogen in Romania, a Central Eastern European country. It highlights the characteristics of the invasive *S. suis* infection, which has a prolonged course and disabling sequelae. Due to the flourishing porcine industry, this pathogen could emerge in Romania as well as throughout Europe.

## 2. Materials and Methods

We carried out a retrospective study by searching the medical electronic database of our institute, an infectious diseases tertiary hospital in Bucharest, the capital of Romania, for all patients admitted between January 2001 and June 2024 who had the final diagnosis of *Streptococcus suis* infection.

All confirmed *Streptococcus suis* cases were collected through the hospital’s electronic database and reviewed.

The etiological diagnosis was established based on at least one positive culture of Streptococcus suis in blood or cerebrospinal fluid (CSF).

In our institute, blood cultures (Bactalert, Biomerieux) are taken in case of signs or symptoms of systemic infection.

The CSF analysis includes cellularity (the number of nucleated cells per cubic millimeter (mm^3^); normal values are less than 5 cells/mm^3^; the percentage of mononuclear cells (lymphocytes and monocytes) and polynuclear cells (mainly neutrophils), with Giemsa smear examination used for further detail. Gram staining is applied to smears of centrifuged samples to identify Gram-positive or Gram-negative bacteria. CSF is cultivated on standard bacterial media: blood agar, chocolate agar. The biochemical examination of glucose, protein, and lactic acid levels in the CSF is also routinely performed, with normal values being greater than 40 mg/dL or greater than half of the concomitant blood glucose level for glucose, 20–60 mg/dL for protein, and less than 18 mg/dL for lactic acid.

We collected and analyzed data regarding demographics, medical history, clinical presentation, occupational risk factors (including pig/pork exposure), blood tests, lumbar puncture results, antimicrobial susceptibility, antibiotics and adjuvant therapies, evolution, outcomes, and complications.

The study received the approval of the Ethics Committee of the National Institute for Infectious Diseases “Prof. Dr. Matei Bals” no 4234/22.04.2024. There was no need for informed consent, due to the retrospective nature of the study. Patient names, personal and other traceable personal information that could be used to identify the patients was omitted and treated as confidential in all processes of data collection and management. The last case (2023) was in the direct medical care of the first four authors of the article.

## 3. Results

Our study identified eight patients admitted during the study period (2001–2023). Between 2001 and 2008, there was no admission. The first patient was admitted in 2008, and the other seven were admitted after 2013, with approximately one case annually (see Table 1), and a pause during the COVID-19 pandemic. Table 1 summarizes the demographical, clinical, and paraclinical characteristics, the risk factors, the treatment, and the outcome of the patients included in this study.

### 3.1. The Year of Admission

Case number: #1:2008; #2:2013, #3:2014, #4: 2015, #5:2016, #6 and #7: 2018, #8:2023

### 3.2. Demographic Features

The median age was 53.3 years (range 42–74), and most patients were male—with a male-to-female ratio of 3:1. Six patients lived in an urban area, and two lived in the countryside.

### 3.3. Epidemiologic Link/Risk Factors and Comorbidities

Risk factors/epidemiological links for acquiring the infection, such as contact with domestic pigs or wild boars, or handling raw pork, were highlighted for three patients: one patient had direct contact with pigs and wild boars (he worked as a forest ranger), and the other two patients had handled fresh pork bought from the local butcher shop.Comorbidities were reported for three out of eight patients: one case of multiple myeloma in immunosuppressive therapy, one case had immunosuppressive therapy (without a mentioned comorbid condition) and one case had important cardiovascular comorbidities: congestive heart failure, ischemic heart disease, intermittent claudication, hypertension, and dyslipidemia.

### 3.4. Clinical Features/Lab Findings/Diagnosis

#### 3.4.1. Diagnosis

The diagnosis at admission was meningitis in seven cases, and in one case, it was endocarditis. During hospitalization, an additional case of endocarditis was found among the seven patients initially diagnosed with meningitis, bringing the total to two patients with endocarditis. Sepsis was associated in two cases (see Table 1)

#### 3.4.2. Clinical Signs and Symptoms

The most frequent symptom at admission was fever associated or not with chills in six out of eight cases (75%). Headaches and nausea associated with vomiting were each reported by four patients. Altered consciousness was the form of presentation for three patients. Due to the severity of illness and sepsis criteria at presentation, two patients were admitted directly to the intensive care unit (ICU); neither required mechanical ventilation. Sudden onset of hearing loss, an important sign of *Streptococcus suis* infection, was present in three cases at admission. One patient had non-specific complaints like irritative cough, photophobia, and myo-arthralgia.

**Table 1 microorganisms-13-00335-t001:** The patients with invasive *S. suis* infection—synoptic panel.

No# Sex Age Dx Year	Living	Risk Factors	Symptoms Admission	Clinical sdr	WBC/mm^3^	Ne/ mm^3^	Ly/ mm^3^	Plt ×10^3^ /mm^3^	Fbg mg/dL	CRP (mg/L)	PCT (μg/L)	Culture (+)	CSF WBC/mm^3^	CSF Proteins (mg/dL)	CSF Lactic Acid (mg/dL)	CSF Glucose/Blood Glucose (mg/dL)	ICU (days)	Antibiotic (days)	Steroids (days)	Sequelae	Hosp Stay (days)
#1 **M 57** 2008	U	NA	fever headache vomiting	Menin gitis	16,200	13,700	1300	178	941	NA	NA	CSF	1400	261	NA	11/113	0	12	28	No	14
#2 **F 45** 2013	U	NA	fever nausea vomiting altered general state (ICU admission)	Menin gitis Sepsis	19,300	18,700	400	89	442	NA	4.98	CSF	1310	155	89	34/155	49	48	26	Hearing loss	49
#3 **M 81** 2014	U	pork	fever myalgia arthralgia confusion	Menin gitis	13,310	11,280	1110	156	651	NA	17.42	CSF	2320	861	NA	20/106	0	15	9	No	19
#4 **F 79** 2015	U	pork	fever vomiting poor general condition	Menin gitis Endo carditis	4450	3650	570	197	649	276	15.72	Blood	1240	148	58.1	28/115	0	43	7	Ataxia	48
#5 **M 58** 2016	U	NA	fever headaches sleepiness hearing loss	Menin gitis	23,100	21,800	700	106	652	281	21.23	CSF	4808	1041	119.2	20/107	0	15	4	Hearing loss Ataxia	16
#6 **M 40** 2018	U	NA	irritating cough decrease exercise tolerance	Endo carditis	17,080	13,910	2360	565	586	76	<0.05	Blood	NA	NA	NA	NA/109	0	30	0	No	30
#7 **M 41** 2018	R	NA	fever headache vomiting hearing loss	Menin gitis	21,800	20,600	600	930	700	4.1	NA	CSF	4800	562	141	20/164	0	14	14	Hearing loss	13
#8 **M 50** 2023	R	pork pigs	headache photophobia altered mental state sudden deafness (ICU admission)	Menin gitis Sepsis	24,860	22,660	540	225	774	292	4.77	CSF	1500	675	135.9	20/145	4	106	18	Deafness (total, bilateral) Ataxia	82

**Legend**: Crt# = current number; M = male; F = female; Age (in years); Dx year = diagnosis year (the year of admission); Living= living conditions: U = urban; R = rural; Risk factors: handling pork or contact with pigs; WBC = white blood cells; Ne = neutrophils; Ly = lymphocytes; Plt = platelets; CRP = C reactive protein; PCT = procalcitonin; CSF = cerebrospinal fluid (!there are listed the findings of the first lumbar puncture); ICU = intensive care unit (days of stay); Hosp = hospital (days of stay).

#### 3.4.3. Laboratory Findings

The median leukocyte cell count at admission was 18.190 cells/mm^3^ (IQR 14.032–22.275)—seven patients out of eight had an increased leukocyte cell count—and the median value for hemoglobin was 12.7 g/dl (IQR 11.8–22.77). Six patients had a low platelet count, and the median value for all eight patients was 167,000 /mm^3^ (IQR 96.250–217.925).Procalcitonin levels were assessed in six patients, with a median level of 10.35 ng/mL (range 3.6–18)—see Table 1.Lumbar punctures (LP) were performed in all seven patients who were admitted with the diagnosis of meningitis. The admission LP (Table 1) had the following characteristics: the median number of cells in the cerebrospinal fluid (CSF) was 1500 cells/mm^3^ (range 1310–4800). All seven patients had a high level of proteins in the CSF—median 562 mg/dL (range 155–861)—reflecting high inflammation and all had a ratio of CSF glucose/ blood glucose below 50%, with a median value of 20 mg/dL (range 20–28). The level of lactic acid in the CSF, determined for five patients, had high levels—the median value was 119.2 mg/dL (range 73.55–138.45). All patients required at least a second LP for control, the median LP number being 4 (range 2–11). Five patients had an LP just before discharge, and all five had abnormal CSF, with a median cell count of 12 cells/mm3 (range 3.5–57) and a median level of proteins of 71 mg/dL (range 48–90).In six out of eight cases, *Streptococcus suis* was isolated from CSF, and in two cases from blood. The isolates were not uniformly tested for antibiotic susceptibility because of the long study period (between 2008, when the first case was diagnosed, and 2023). Seven out of seven tested for penicillin were susceptible, and six out of six tested for the third-generation cephalosporins and glycopeptides were susceptible. Two out of seven tested for lincosamides proved to be susceptible. Only two isolates were tested for macrolides and were resistant.

### 3.5. Treatment

#### 3.5.1. Antibiotic Therapy

All therapeutic regimens included at least a beta-lactam, third-generation cephalosporins being the most used antibiotic (see Table 2).Six out of eight patients received at least two regimens of antibiotics, the median number of different regimens being 2.5 (range 1–8). The most common association was with vancomycin (*n* = 4), linezolid (*n* = 2), and gentamycin (*n* = 2). Both patients with endocarditis were treated with regimens that included ceftriaxone, ampicillin, and gentamycin, and both patients admitted to the ICU received meropenem in association with vancomycin or linezolid. The average duration of the antibiotic therapy was 36 days (median 22.5, range 12–106 days), and three patients had a total duration of antimicrobial therapy of more than 42 days. The patient with recrudescent meningitis received a total of 106 days of antibiotics.

#### 3.5.2. Corticosteroids

Adjuvant corticosteroid therapy with dexamethasone was used in all seven cases with meningitis, for a median duration of 11.5 days (range 5–24) (see Table 1).

#### 3.5.3. Intensive Care Treatment

Two patients were admitted directly to the ICU for intense care and support, but neither of them required mechanical ventilation. One of them (#8 in Table 1 and Table 2) was transferred after 4 days to the regular yard, and the other (#2 in Table 1 and Table 2) stayed for 49 days and was then transferred to another setting for chronic care and rehabilitation.

### 3.6. Evolution and Outcome

The median number of days of hospitalization was 24.5 (IQR 14; 49) (range 13–82).The ICU admission needed for two patients was previously described (Section 3.5.3).Our patients had prolonged antibiotic and steroid therapy (with a median of 22.5 days and 11.5 days, respectively—see Section 3.5.1 and Section 3.5.2, respectively);The subsequent CSF analyses (at least a second one during hospitalization, with a median of four lumbar taps) showed improvement but not total normalization at discharge (see Section 3.4.3).During hospitalization, an additional patient developed hearing loss (resulting in a total of four cases). In three cases, it was partial (hypoacusis) with slight recovery and in one case was total, bilateral and not reversible.Ataxia was recorded in three cases during hospitalization, with partial recovery at discharge.In two cases, the echocardiogram revealed cardiac vegetation, and one of these two patients also presented concomitant meningitis.No deaths have been reported, but five of our patients remained with sequelae: four had various degrees of hearing loss with irreversible profound bilateral deafness in one case. Two of them also developed vestibular dysfunction (ataxia), and there was an additional patient who developed ataxia without having meningitis or hearing loss. (see Table 1); the patients with ataxia almost fully recovered. There was one case (#8) with relapse—the patient was readmitted to the hospital and received a prolonged antibiotic course (of 106 days in total). He received oral antibiotics between the first and the second hospital stays.

## 4. Discussion

Our study found roughly one admission per year in a Romanian tertiary infectious diseases hospital since 2008, except for the COVID-19 pandemic. The median age of 53 years, and the male predominance (male-to-female ratio of 3:1) are similar to those reported by most studies.

Regarding the risk factors, the reports in Western Europe [2,3,60] are about sporadic cases and a high percentage of missing epidemiological links. However, a systematic review and meta-analysis of studies performed in all world regions, published in 2015 by van Samkar et al., reported that 61% of patients had contact with pigs or pork [61]. In our study, we found that one-third of our patients handled pork and/or pigs, but given that our study is retrospective and concerns a non-reportable disease in Europe, it is more likely to have missing data related to risk factors rather than an actual absence of these factors.

Meningitis is the most common form of presentation regardless of the world region, followed by sepsis—including toxic shock syndrome with multiple organ dysfunction syndrome, septic arthritis, endocarditis, and ophthalmitis.

Most reviews report that more than half of the patients (50–60%) have meningitis at admission [2,52], as in our study—with seven out of eight patients presented with meningitis. Additionally, there were no significant differences regarding clinical presentation and outcome of meningitis between different studies or even reviews/meta-analyses, geographical regions of the world, or even between low and high-income countries [1,2,3,43,61,62]. Most studies report that the sensitivity of the classic triad of bacterial meningitis signs—fever, altered mental status, and neck stiffness—is low in *Streptococcus suis* infection [61]. Similarly, in our study, fever was present in just two-thirds of the cases, headache and nausea were reported by half, and altered consciousness was present in one-third of cases. Self-reported sudden onset of hearing loss was reported at admission in one-third of cases. There are a few hypotheses for the mechanism responsible for hearing loss, including direct infection of the auditory nerve and labyrinthitis (suppurative or hemorrhagic) [61,66] similar to other bacterial meningitis [67]. Hearing loss represents the most common sequelae reported after *Streptococcus suis* meningitis. In our study, half of the patients (four out of eight) had hearing loss, with one patient (the one diagnosed in 2023) having total bilateral hearing loss from the disease’s onset, and it was irreversible despite the very early antibiotic and corticosteroid treatment. A review and meta-analysis published by Edmond et al. looked at global and regional risk of disabling sequelae from all kinds of bacterial meningitis. The authors reviewed 132 papers published from 1980 to 2008. They classified the sequelae into two categories: major (cognitive deficit, bilateral hearing loss, motor deficit, seizures, visual impairments, and hydrocephalus) and less severe, minor (behavioral problems, learning difficulties, unilateral hearing loss, hypotonia, diplopia). Pneumococcal meningitis was associated with the greatest risk of major sequelae at 24.7%. The most common sequela was hearing loss (33.95%), and the risk of long-term important sequelae was the highest in low-income countries from Africa and South-East Asia [67]. Another study by Jensen et al. found that more than 60% of patients diagnosed with acute bacterial meningitis were affected by hearing loss despite treatment with dexamethasone [68]. Hearing loss is the most common complication for *Streptococcus suis* meningitis in humans, but the reported rates of this complication vary greatly in the literature [50,69,70]. For example, a review by Hughes et al. published in 2009 reported that 54% up to 68.8% of patients had some degree of hearing loss at discharge. Some patients improve over time, while most do not. The same authors reported that 66.4% of 93 Vietnamese adults had mild-to-severe hearing loss at hospital discharge compared to 47.7% of 86 patients evaluated 6 months later [50,71]. Another study that included 44 Vietnamese patients reported profound-to-complete and moderate-to-severe hearing impairments in 32% (*n* = 14/44) and 52% (*n* = 23/44) of cases at discharge, respectively. The percentage fell to 29% in both groups at the 9-month evaluation, and the study concluded that most improvement was observed in patients with non-severe hearing loss [70]. Ataxia (vestibular dysfunction), another complication, can be associated with hearing loss [72] and studies have found that it occurs more often in patients with severe to complete hearing loss [73]. Huong et al. found that similar to hearing loss, ataxia would persist in most cases at the 9-month evaluation, meaning a definitive permanent loss of function [70]. In our patients, two had ataxia, reversible in both cases.

*S. suis* endocarditis occurs less often than meningitis [74]. When endocardial infection occurs, because of bacteria`s propensity for biofilm formation [75], significant valvular destruction, distal embolization, and also perivalvular complications can appear [74,76]. Despite these, there is no recommendation for routine transthoracic echocardiography examination in patients with *Streptococcus suis* bacteremia [77]. We had two patients with endocarditis, one of them having both endocarditis and meningitis.

The blood tests of our study patients were characteristic of bacterial infection: high leukocyte cell count with high neutrophils and an important inflammatory response, including a high level of procalcitonin. Procalcitonin is an important marker for invasive bacterial infections, and most experts agree that the procalcitonin level correlates with the severity of infection [78]. In parallel, the CSF changes are characteristic of bacterial meningitis: high neutrophil cell count, high protein and lactic acid levels, and low glucose levels. These findings are similar to those reported by other studies from all over the world [50,52,60,61,62].

In our cohort, most specimens (75%; *n* = 6) were isolated from the CSF, and only 25% (*n* = 2) from blood samples. All bacterial isolates tested were susceptible to beta-lactam antibiotics (including penicillin) and glycopeptides. On the other hand, *Streptococcus suis* presented high levels of resistance to lincosamides and macrolides, like those reported from studies in both humans and pigs, including animal studies from Romania [37,38,39,40]. All patients with meningitis were started on a regimen containing a third-generation cephalosporin (ceftriaxone) and vancomycin. The regimen was adjusted once the susceptibility test was available. However, despite general susceptibility to penicillin, none of the patients received a penicillin-containing regimen. More than that, the two patients admitted to the ICU received carbapenems in association with a glycopeptide or linezolid. One explanation for this situation could be that the changes in the CSF were persistent despite correct antibiotic treatment and the association of dexamethasone. Additionally, the patient who experienced the relapse of meningitis received oral aminopenicillin at discharge and experienced a worsening of symptoms while taking this treatment. The consensus on *Streptococcus suis* infection treatment in most studies is that patients need prolonged antibiotic treatment, although an exact duration was not established. Some authors report relapses even after four weeks of antibiotic therapy [50,71,79,80,81,82,83]. A systematic review published by Rayanakorn et al. in 2018 established that the antibiotic treatment duration ranged from 7 to 42 days, while the reported forms of the disease were meningitis, spondylitis, and endocarditis [46]. In our study, the average duration of the antibiotic therapy was 36 days, and the patient with recrudescent meningitis received a total of 106 days of antibiotics. In the literature, the relapse rate is low. Data published by Huong et al. in 2014 showed that the relapse rate was 4.4% after cessation of antimicrobial drug treatment, even in patients for whom the microorganism tested highly susceptible to penicillin [52], and this was the case in our last patient included in the study.

Guidelines recommend the use of corticosteroids (more precisely, dexamethasone) as an adjunctive treatment for bacterial meningitis as it improves the outcome; it reduces hearing loss and other neurologic sequelae, and it can reduce mortality, proven for *Streptococcus pneumoniae* meningitis [84,85]. Regarding *Streptococcus suis* infection and meningitis, dexamethasone is usually used, as some randomized controlled trials have shown that corticosteroids reduce hearing loss in at least one ear and reduce the risk of severe hearing loss [61,71]. However, there is still controversy, as other authors have reported a high prevalence of hearing loss despite dexamethasone treatment [62]. All patients with diagnosed meningitis in our study group had dexamethasone associated with the antibiotic treatment, for a median duration of 11.5 days. Despite this fact, the prevalence of hearing loss at discharge was 50%.

The mortality in *Streptococcus suis* infection is low and depends on the localization and presence or absence of sepsis. Most authors give a mortality rate of around 3–4% for *Streptococcus suis* meningitis and 10–13% for general invasive infection, especially in the presence of streptococcal toxic shock syndrome [61,86]. Mortality was higher in patients from outbreaks like those reported from China and Thailand [47,56,57]. We had no deaths in our group.

Our study has limitations, mainly because it is a retrospective one and included a small number of patients. *S. suis* is probably an underdiagnosed etiological agent in our country, considering the present risk factors (emergence of pig farming in individual households, among others). Raising awareness within the medical community (human medicine in collaboration with veterinary medicine) could significantly improve both the diagnosis and prognosis of these serious infections. We proposed a ’One Health’ approach to this issue in Romania, as a future goal. Another limitation is the unavailability of serotype testing to determine what serotypes are circulating in Romania. These could also be important issues for future studies that aim to characterize *S. suis* infections in Romania and across Europe. These could also be important issues for future studies that aim to characterize *S. suis* infections in Romania and across Europe, to probe the possible link between the *S. suis* serotypes 2 and much more frequent and serious illnesses in humans and animals [1,19,20].

## 5. Conclusions

This study is the first paper to report a series of cases of *Streptococcus suis* infections from Romania, a European country with a long history of pig raising. The pattern of infections in Romania is similar to that found in the rest of Europe, with sporadic cases mostly occurring in men. Epidemiological links—risk factors such as handling pigs or pork (raw meat)—with the *Streptococcus suis* infection were found in one-third of the patients. The typical clinical presentation of bacterial meningitis, which consists of the classical triad of fever, headache, and neck stiffness, is missing in an important percentage of cases. Instead, *Streptococcus suis* infection is often associated with abrupt onset of hearing loss. This should prompt the clinician to consider this uncommon pathogen, especially in the presence of risk factors like pig or pork contact. With the increasing number of cases diagnosed in the last years and the potential severity of this disease, clinicians and microbiologists need to be aware of *Streptococcus suis* as a human pathogen even in non-endemic areas like Romania.

## Figures and Tables

**Table 2 microorganisms-13-00335-t002:** The antibiotic regimens and the duration of their use in our eight cases.

Cases	Total Duration of Antibiotics	Antibiotics Regimens
#1	12 days	Ceftriaxone (2 g IV q12h) for 12 days
#2	48 days	Ceftriaxone (2 g IV q12h) + Vancomycin (1 g IV q12h) for the first day, then Meropenem (2 g IV q8h) + Vancomycin (1 g IV q12h) for 3 days, then Ceftriaxone (2 g IV q12h) for 15 days, then Linezolid (600 mg IV q12h) for 3 days, then Meropenem (2 g IV q8h) + Linezolid (600 mg IV q12h) for 17 days, then Meropenem (2 g IV q8h) for 9 days
#3	15 days	Ceftriaxone (2 g IV q12h) + Vancomycin (1 g IV q12h) for 4 days, then Ceftriaxone (2 g IV q12h) for 11 days
#4	43 days	Ceftriaxone (2 g IV q12h) + Gentamicin (1 mg/kg q8h) for 18 days, then Ampicillin (3 g IV q8h) for 25 days
#5	15 days	Ceftriaxone (2 g IV q12h) + Vancomycin (1 g IV q12h) for 4 days, then Ceftriaxone (2 g IV q12h) for 11 days
#6	30 days	Ampicillin/sulbactam (4.5 g IV q6h) + Gentamicin (1 mg/kg q8h) for 23 days, then Ceftriaxone (2 g IV q12h) + Ciprofloxacin 7 days
#7	14 days	Ceftriaxone (2 g IV q12h) for 10 days, then Ceftriaxone (2 g IV q24h) for 4 days
#8	106 days	Ceftriaxone (2 g IV q12h) for 13 days (during the first hospitalization) then Amoxicillin (2 g po q8h) for 7 days (at discharge; clinical relapse and readmission), then Ceftriaxone (2 g IV q12h) for 10 days, then Ceftriaxone (2 g IV q12h) + Linezolid (600 mg IV q12h) for 5 days, then Meropenem (2 g IV q8h) + Vancomycin (1 g IV q8h) for 17 days, then Meropenem (2 g IV q8h) for 26 days, then Meropenem (2 g IV q8h) + Rifampicin (450 mg po q12h) for 6 days, then Ampicillin (3 g IV q6h) + Moxifloxacin (400 mg po q24h) for 9 days, then Amoxicillin (2 g po q8h) + Moxifloxacin (400 mg po q24h), for 14 days

Cases from #1 to #8 are listed chronologically in order of their admission year.

## Data Availability

The data presented in this article are available upon request from the corresponding author. The original contributions presented in this study are included in the article. Further inquiries can be directed to the corresponding author. The raw patients’ data are preserved in the electronic database of the National Institute for Infectious Diseases Matei Bals, Bucharest, Romania, and consist of their files (the discharge file containing the whole illness history) and all laboratory tests. Our Institute’s database contains the patients’ data since 2000.

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
