# Peer review of "Streptococcus suis: A Possible Emerging Zoonotic Pathogen in Romania"

_microorganisms, 2025, doi:10.3390/microorganisms13020335_

Round 1

Reviewer 1 Report

Comments and Suggestions for Authors

“There was one case (#8) with relapse – the patient was readmitted to the hospital 103 and received a prolonged antibiotic course (of 82 days in total).”
I suggest you specify whether you are referring to the duration of hospital stay or the duration of antibiotic therapy. This sentence suggests that it is the latter, however, which disagrees with the information on case 8 in the table.

The authors write: "no external funding" but acknowledge institutional support later (Line 352) and additionally provide the name of the scientific program. This sounds unambiguously like funding information. Please clarify and specify.

I suggest ensuring all abbreviations are defined at first use for clarity (e.g. CSF, ICU)

The introduction repeats information about the zoonotic potential of Streptococcus suis and its global distribution without adding something new.

I suggest condensing these parts and focusing more on the unique context of Romania, as highlighted later.

The limitations section mentions the retrospective design and lack of serotyping of the strains obtained but does not suggest how future studies could address these gaps. There should be a brief explanation of why serotyping just this pathogen might be of epidemiological interest, for example.

We write the names of bacteria in italics. Please review the text and correct this error

Is ethics committee approval not required for access to patient data? In the text, it is indicated that it is, but the number/designation of this consent is not given.

Author Response

Dear Reviewer,

Thank you very much for taking the time to review this manuscript and for your beneficial observations and suggestions! We have tried to correct our manuscript accordingly and hope we have succeeded. The corrections are highlighted in grey in the attached manuscript.

Comment #1: “There was one case (#8) with relapse – the patient was readmitted to the hospital 103 and received a prolonged antibiotic course (of 82 days in total).”
I suggest you specify whether you are referring to the duration of hospital stay or the duration of antibiotic therapy. This sentence suggests that it is the latter, however, which disagrees with the information on case 8 in the table.

Response #1: Yes, indeed, our sentence was unclear. Thank you for pointing this out! The patient was hospitalized for a total of 82 days (13 days during the first stay, discharged on request, and readmitted after several days for a second hospitalization of 69 days). Between the two hospitalizations, he received oral antibiotic therapy, which resulted in a total antibiotic therapy duration of 106 days. We corrected this—please see lines 116-117. We also corrected two errors in Table 1 (highlighted in grey) and added serial numbers to the patients in Table 1.

Comment #2 The authors write: "no external funding" but acknowledge institutional support later (Line 352) and additionally provide the name of the scientific program. This sounds unambiguously like funding information. Please clarify and specify.

Response #2: Yes, we had a dilemma regarding the funding status. On one hand, there is the research work for which we have no funds. On the other hand, the University could partially or totally retroactively cover the publication's cost through the program Publish not Perish. We modified this accordingly; please see lines 271-273.

Comment #3 I suggest ensuring all abbreviations are defined at first use for clarity (e.g. CSF, ICU)

 Response #3: We apologize for this absence. We have added the abbreviations in the Methods Section (lines 197-215). Thank you!

Comment #4: The introduction repeats information about the zoonotic potential of Streptococcus suis and its global distribution without adding something new. I suggest condensing these parts and focusing more on the unique context of Romania, as highlighted later.

Response #4: Yes, thank you. The information in both the introduction and discussion sections was redundant. Your point is valid, and we have attempted to group and condense the information. In this regard, we have moved an important part from the discussion section to the introduction section, as also recommended by Reviewer #2. Please see lines 40-154 for Introduction and 135-136, 149-150, 175-176, 181-182 and 215 for Discussions.

Comment #5 The limitations section mentions the retrospective design and lack of serotyping of the strains obtained but does not suggest how future studies could address these gaps. There should be a brief explanation of why serotyping just this pathogen might be of epidemiological interest, for example.

Response #5: Thank you for pointing this out! We suggest a future approach in this regard—please see lines 236-245.

Comment #6 We write the names of bacteria in italics. Please review the text and correct this error

 Response #6: Yes, we corrected this and hope there are no instances of S. suis that are not italicized in the manuscript!

Comment #7 Is ethics committee approval not required for access to patient data? In the text, it is indicated that it is, but the number/designation of this consent is not given.

Response #7: Yes, the Ethics Committee approved access to patients' data, and we have added the number and date of this approval. Thank you! Please see line 190 in the Methods section. 

Thank you once again for your time and help!

Reviewer 2 Report

Comments and Suggestions for Authors

In the reviewed MS a group of scientists from Bucharest (Romania) report on their retrospective study summarizing data on eight patients infected by pathogenic bacterium Streptococcus suis admitted to a hospital in Romania in 2008-2024. The MS is written as a factual report. Distinct goals of the study are not specified in the text. The Introduction is very short and uninformative. It lacks some important information, e.g. on the general data on the effects caused by Streptococcus suis (health problems, pathology) when a human is infected. Methodology is written very briefly and needs better explanation. The Section Results mainly retells the content of a large Table, which gives primary data on the eight analyzed patients. The Discussion is too long and unfocused. It is mainly because the researchers did not specifies the goals of the study and consider their work mainly as a medicinal descriptive report. First half of Discussion is redundant, this content in a brief form could be given in the Introduction. The authors are requested to provide some graphics and diagrams illustrating their results and give a full list of abbreviations in the section Material and methods. The MS needs serious, complete revision. It also needs some linguistic corrections. Some additional remarks are given below.

20 Streptococcus suis - italic

22 over 24 years (2001-2024) – repetition, 2001-2024 is enough

29 half of our patients experienced severe sequelae – please, specify if it was because of Streptococcus suis or not, not clear

30-32 – suboptimal sentence, Please consider revising

36 Streptococcus suis belongs to a family of Gram-positive bacterial strains – which family? Please, give the distinct taxonomy

36-40 please provide references for the information in this paragraph

45 fatal sepsis- remove “-”

46-47 Most cases originated from Asia (90.2%) while human infections in Europe account for approximately 8.5% of the global prevalence. There were only eight (0.5%) cases of human infections reported in North America [6]. – this is too long, please, say this shorter, e.g.

Most cases originated from Asia (90%), and only 8.5% and 0.5% in Europe and North America respectively.

Most cases originated from Asia (90%), notably more than in Europe (8.5%) and North America (0.5%).

Introduction: please add the general information of the effects caused by Streptococcus suis (health problems, pathology) when a human is infected.

70 We collected data regarding – please specify distinctly the source of your data: archives, journal, disease history of patients etc

73 Our definition of Streptococcus suis infection included isolating the microorganism from a normally sterile site in a patient with symptoms and signs of infection. – this needs better explanation. Please write a paragraph explaining what exactly has been done

Material and Methods – please distinctly explain if your study is baes only on retrospective data of some other researchers or it also includes your own experimental work

80 2001-2023 – please compare with Abstract

Results: please give a list of all abbreviations, preferably in the section Material and Methods

Discussion: 106-210 all this text report on literature data which should be given in the first part of the paper, in the Introduction.

Discussion is unfocused and too long.

Author Response

For research article

Response to Reviewer 2 Comments

Dear Reviewer,

Thank you very much for taking the time to review this manuscript and for your helpful input! We have tried to correct it accordingly and hope we have succeeded. The corrections are highlighted in grey in the attached manuscript.

Comment #1: In the reviewed MS a group of scientists from Bucharest (Romania) report on their retrospective study summarizing data on eight patients infected by pathogenic bacterium Streptococcus suis admitted to a hospital in Romania in 2008-2024. The MS is written as a factual report. Distinct goals of the study are not specified in the text. The Introduction is very short and uninformative. It lacks some important information, e.g. on the general data on the effects caused by Streptococcus suis (health problems, pathology) when a human is infected. Methodology is written very briefly and needs better explanation. The Section Results mainly retells the content of a large Table, which gives primary data on the eight analyzed patients. The Discussion is too long and unfocused. It is mainly because the researchers did not specifies the goals of the study and consider their work mainly as a medicinal descriptive report. First half of Discussion is redundant, this content in a brief form could be given in the Introduction. The authors are requested to provide some graphics and diagrams illustrating their results and give a full list of abbreviations in the section Material and methods. The MS needs serious, complete revision. It also needs some linguistic corrections. Some additional remarks are given below.

Response #2: We agree and thank you for your helpful and constructive comments. Indeed, there is a retrospective description of eight patients admitted to our Institute – the largest tertiary infectious diseases hospital in Romania – over a very long period. The first case was recorded in 2008, even though our search in the electronic medical database started in 2001. The patients had severe infections with Streptococcus suis, a zoonotic agent affecting humans and pigs. In the introduction, we added the aim of our study, which is mainly to raise awareness about this serious pathogen that seems to be emerging due to the flourishing porcine industry and pork consumption (please see lines 160-163 in the Introduction section).

Indeed, the Introduction was very short and uninformative, and we have improved it following your suggestion. We added general data about the bacterium and human disease and moved important information from the Discussion section here (please see lines 40-154 for the Introduction).

The Methodology section was extended and improved, with an abbreviation column added. The Results section was also improved, mainly describing the patients' data in subsections 3.1-3.6. (please see lines 1-69 and 73-117 in the Results section)

As you suggested, the Discussion section was condensed, and roughly half was moved to the Introduction section. Consequently, the Discussion section was shortened, and its content is now more specific and related to our study's results.

Comment #2:20 Streptococcus suis - italic

Response #2: Yes, we corrected this and hope there are no instances of S. suis that are not italicized in the manuscript!

Comment #3: 22 over 24 years (2001-2024) – repetition, 2001-2024 is enough

Response #3: we corrected accordingly – please see line 22

Comment #4: 29 half of our patients experienced severe sequelae – please, specify if it was because of Streptococcus suis or not, not clear

Response #4: Yes, the sequelae are related to Streptococcus suis infection, and we try to clarify this aspect in the text – please see lines 29-30.

Comment #5: 30-32 – suboptimal sentence, Please consider revising

Response #5: Indeed the sentence is uninformative, so we replaced it – please see lines 31-32.

Comment #6: 36 Streptococcus suis belongs to a family of Gram-positive bacterial strains – which family? Please, give the distinct taxonomy

Response #6: Thank you for the comment. We added the requested data—please see lines 36-37.

Comment #7 :36-40 please provide references for the information in this paragraph

Response #7: We added the references [1] and [2] for the information in the mentioned paragraph (please see lines 38 and 39.

Comment #8: 45 fatal sepsis- remove “-”

Response #8 We removed “-“ (line 115 in Introduction)

Comment #9: 46-47 Most cases originated from Asia (90.2%) while human infections in Europe account for approximately 8.5% of the global prevalence. There were only eight (0.5%) cases of human infections reported in North America [6]. – this is too long, please, say this shorter, e.g.

Most cases originated from Asia (90%), and only 8.5% and 0.5% in Europe and North America respectively.

Most cases originated from Asia (90%), notably more than in Europe (8.5%) and North America (0.5%).

Response #9: Thank you. We modified the following as your suggestion: ‘Most cases originated from Asia (90%), notably more than in Europe (8.5%) and North America (0.5%)’—please see lines 116-117 in the Introduction.

Comment #10: Introduction: please add the general information of the effects caused by Streptococcus suis (health problems, pathology) when a human is infected.

Response #10: In the Introduction section, we introduced distinct paragraphs about S. suis in pigs and the infection in humans—please see lines 109-154, but we are more focused on lines 109-113.

Comment #11: 70 We collected data regarding – please specify distinctly the source of your data: archives, journal, disease history of patients etc

Response #11: We collected data from our electronic medical database at the National Institute for Infectious Diseases, where four of us have been working. The last patient described in this work—the most severe one—was in our direct care, and the idea for this manuscript emerged from him (please see lines 193-194 in the Methods section). His case raised many questions about the prolonged course, serious and irreversible sequelae, and relapse despite antibiotics that were proven to be efficacious. In the section Methods, we clarified from where we collected our data: lines 165-170.

Comment #12: 73 Our definition of Streptococcus suis infection included isolating the microorganism from a normally sterile site in a patient with symptoms and signs of infection. – this needs better explanation. Please write a paragraph explaining what exactly has been done

Response #12: We agree that it was not clear in the Method section, so we rephased it: please see lines 171-183

Comment #13: Material and Methods – please distinctly explain if your study is baes only on retrospective data of some other researchers or it also includes your own experimental work

Response #13: Thank you for the comment! We tried to clarify in the Methods section, adding information about our study: it is a retrospective descriptive study that analyzed the patients with the final diagnosis of S. suis infection admitted to the National Institute for Infectious Diseases, where four of us have been working (as we specified in our response #11). Also specified in our response #11, the last patient described in this work—the most severe one—was in our direct care (please see lines 193-194 in the Methods section). The other seven patients were in other colleagues’ care, in the same setting. There is no experimental work; we are physicians and here is a description of clinical work with our patients in care. In the section Methods, we clarified from where we collected our data: lines 165-187. We had the Institute Ethics Committee's approval to access the Institute's medical database (please see lines 189-194).

Comment #14: 80 2001-2023 – please compare with Abstract

Response #14 Thank you for pointing this out; our sentences were unclear. Our search in the electronic database is addressed to the period 2001-2024, but the first case was admitted with the final diagnosis of an S. suis infection dated 2008. Hope we clarified this aspect in lines 217-218 in the Results section.

Comment #15: Results: please give a list of all abbreviations, preferably in the section Material and Methods

Response #15: We apologize for this absence. In the Methods Section (lines 197-215), we have added a column with the abbreviations.

Comment #16: Discussion: 106-210 all this text report on literature data which should be given in the first part of the paper, in the Introduction section.

Response #16: Thank you for this very valuable suggestion! Indeed, the manuscript is now more balanced. We have moved the text from the Discussion section to the Introduction section – please see lines 40-154 in the Introduction, as you recommended.  

Comment #17: Discussion is unfocused and too long.

Response #17: Yes, indeed, there was too much and somehow redundant information in the Discussion section. Moving its first half (as you recommended, and we are grateful for this!) to the Introduction section, the Discussion section was shortened, and its content is now more specific and related to the results of our study. We added information about our cases in lines 135-136, 149-150, 175-176, 181-182, and 215 and about our possible future studies in lines 236-245 in the Discussions section.

Thank you once again for your time and help!

Round 2

Reviewer 1 Report

Comments and Suggestions for Authors

None.
The authors have responded to my comments.